

# Immune function of miR-214 and its application prospects as molecular marker

Qiuyuan Wang, Yang Liu, Yiru Wu, Jie Wen and Chaolai Man

College of Life Science and Technology, Harbin Normal University, Harbin, China

## ABSTRACT

MicroRNAs are a class of evolutionary conserved non-coding small RNAs that play key regulatory roles at the post-transcriptional level. In recent years, studies have shown that miR-214 plays an important role in regulating several biological processes such as cell proliferation and differentiation, tumorigenesis, inflammation and immunity, and it has become a hotspot in the miRNA field. In this review, the regulatory functions of miR-214 in the proliferation, differentiation and functional activities of immune-related cells, such as dendritic cells, T cells and NK cells, were briefly reviewed. Also, the mechanisms of miR-214 involved in tumor immunity, inflammatory regulation and antivirus were discussed. Finally, the value and application prospects of miR-214 as a molecular marker in inflammation and tumor related diseases were analyzed briefly. We hope it can provide reference for further study on the mechanism and application of miR-214.

## INTRODUCTION

MicroRNAs (miRNAs) are a kind of high conserved non-coding small RNAs in evolution that bind to the 3′-untranslated region (3′-UTR) of target gene mRNA and regulate gene expression at post-transcriptional level. In immune responses, miRNAs act as signal-regulating molecules after immune-related receptors activation, and affect the expression of immune-related genes, thus extensively participating in various aspects of immune response (*Bosisio et al., 2019*; *Mehta & Baltimore, 2016*).

MiR-214, one of the key miRNAs involved in immune response, is widely distributed in fish, amphibians, birds, mammals and other vertebrates (*Thomas, Adam & John, 2014*). Hsa-miR-214 is located in the intron of *dynamin-3* gene and has-miR-199a is located about 6 kb next to miR-214, and miR-199a/miR-214 cluster often participates in regulating the same reactions. For example, miR-199a/miR-214 cluster can target *E-cadherin* and *claudin-2* and promote high glucose-induced peritoneal fibrosis (*Lee et al., 2009*; *Che et al., 2017*). In human, pre-miR-214 can encode two mature miRNAs: miR-214-5p and miR-214-3p, and miR-214-5p is hardly expressed, while miR-214-3p is high expressed based on 136 published RNA-seq experiments (http://www.mirbase.org/cgi-bin/mirna_entry.pl?acc=MI0000290), so there are functional differences between them (*Bartel, 2004*; *Teng, Ji & Zhao, 2018*; *Li, Wang & Ren, 2018*; *Deng et al., 2019*; *Wang et al., 2019a*; *Wang et al., 2019b*) (Fig. 1).

Corresponding author
Chaolai Man, manchaolai@126.com

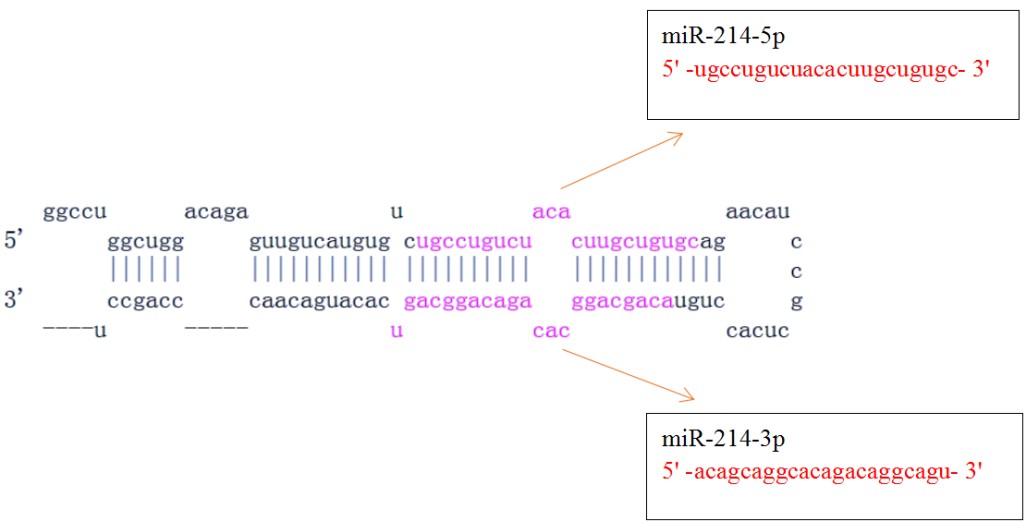

**Figure 1** **Pre-miR-214 and its mature miRNAs (miR-214-5p and miR-214-3p).** (http://www.mirbase.org/cgi-bin/mirna_entry.pl?acc=MI0000290).

Recently, a number of studies have reported the function and mechanism of miR-214 in the fields of immune response, tumor, cardiovascular, development, neurology, and metabolism, which has been a hotspot of miRNAs study. This review mainly focuses on the immune functions and application prospects of miR-214.

## Survey methodology

We used the PubMed database to search the keywords "miR-214", combined with "immunity", "inflammation", "T cell", "tumor immunity", "virus", "molecular marker" to obtain relevant articles and summarized them. Among them, miR-214 combined with "inflammation" retrieved 54 articles, combined with "immunity" retrieved 39 articles, combined with "T cell" retrieved eight articles, and "tumor immunity" retrieved 15 articles, combined with "virus" retrieved 33 articles. Finally, in these articles, we focused on screening the documents directly related to miR-214, and removed the non-key studies on miR-214 and documents not directly related to miR-214. The search was conducted in December 2018, a repeated search was conducted in October 2019, and a third repeated search was conducted in August 2020. The final reference time of our manuscript was from 1998 to 2020.

## MiR-214 and immune cells

MiR-214 regulates the functions and characteristics of a variety of immune cells including dendritic cells (DCs), T cells, natural killer (NK) cells, and macrophages, etc., and participates in immune response processes widely.

MiR-214 is a key miRNA that regulates the functions of DCs. Studies have found that DCs immune activity is inhibited by regulatory T (Treg) cells, and the down-regulation of miR-214-3p can enhance the expression of heat shock protein 27 (HSP27) which inhibits the differentiation of Treg cells, so the overexpression of miR-214-3p can promote the
**A**

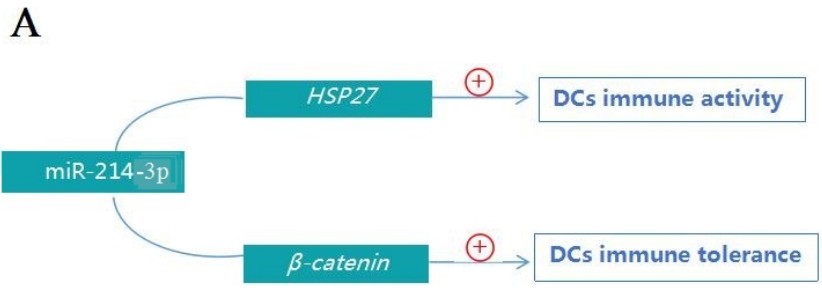

**B**

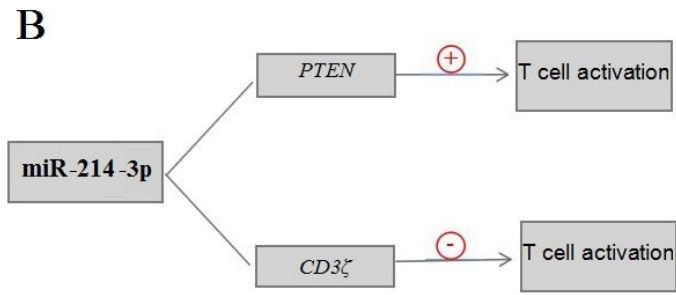

**Figure 2  Pathways of miR-214-3p in regulating the functions of DCs and T cells.** (A) The pathways of miR-214-3p in regulating DCs functions; (B) the pathways of miR-214-3p in regulating T cells activities; ⊕ indicates promotion, and ⊖ indicates inhibition.

differentiation of Treg cells and enhance its inhibitory effect on DCs immune activity (*Pandey et al., 2013*; *Alexander & Steffen, 2014*; *Svajger & Rozman, 2014*; *Huan et al., 2017*). Moreover, miR-214-3p can target the 3′UTR of the *β-catenin* which is a key regulator of DCs tolerance, down-regulation of miR-214-3p can induce DCs immune tolerance (*Fändrich, 2010*; *Gordon et al., 2014*; *Gu et al., 2015*). Therefore, miR-214-3p comprehensively affects the immune activity and tolerance of DCs by regulating the expression of *β-catenin* and differentiation of Treg cells (Fig. 2). In addition, tolerogenic DCs can promote central or peripheral tolerance through the deletion of T cells, induction of Tregs, expression of immunomodulatory molecules, and the production of immunosuppressive factors (*Ilarregui et al., 2009*; *Ezzelarab & Thomson, 2011*; *Li & Shi, 2015*), so miR-214 may have potential application value in the rejection of organ transplantation and the prevention and treatment of autoimmune diseases.

MiR-214 plays an important regulatory role in cellular immune response by regulating the activation, proliferation and differentiation of T cells. For example, PTEN protein, an inhibitor of the PI3K-AKT signaling pathway, negatively regulates T cell activation (*Buckler, Liu & Turka, 2008*). MiR-214-3p can target PTEN and improve the activity of PI3K-AKT signaling pathway, which promotes T cells activation (*Jindra et al., 2010*). In addition, the up-regulation of miR-214-3p in activated T cells can enhance the proliferation capacity
of T cells (*Jindra et al., 2010*). Interestingly, miR-214-3p also inhibits T cell activation because T cell activation requires the signal integration and transduction by T cell receptor (TCR)-CD3 complex, and CD3$\zeta$ plays a key role in the signal transduction. MiR-214-3p can target the 3′-UTR of CD3$\zeta$ gene and negatively regulate T cell activation (*Xiao et al., 2019*). Therefore, miR-214-3p plays a key regulatory role in balancing the activation of T cells (Fig. 2). In addition, gga-miR-214 is significantly up-regulated in chicken thymus under immunosuppressive condition, but whether the up-regulation of gga-miR-214 is related to T cell suppression in thymus is worthy of further study (*Zhou et al., 2019*). It is worth mentioning that miR-214 also regulates the differentiation of certain T cells. MiR-214 plays a key role in the differentiation of naive T cells to Th17 cells during the relapse and remission phases of multiple sclerosis (MS) patients, because miR-214 may be a negative regulator of Th17 cell differentiation and the imbalance of Th17 cells is a key factor leading to MS (*Brucklacher-Waldert et al., 2009*). Therefore, miR-214 may have potential value in the prevention and treatment of MS (*Ahmadian-Elmi et al., 2016*).

MiR-214 plays an important role in regulating functions of NK cells and macrophages. NK cells are closely related to the occurrence and maintenance of pregnancy (*Bezman et al., 2010*; *Nabila, 2019*). MiR-214 is differentially expressed between uterus decidual and peripheral blood natural killer cells, which may affect the NK cells differentiation and the process of pregnancy (*Carlino et al., 2018*). So, in-depth research on miR-214 regulating NK cells functions can help to understand the pathological mechanisms of pregnancy injury at molecular level. It is well known that macrophages are important immune cells with phagocytic function and mediate the transition from innate immunity to adaptive immunity (*Arango Duque & Descoteaux, 2014*). Study shows that virus mimics polyriboinosinic polyribocytidylic acid (pIC) can induce up-regulation of 10 miRNAs expression in Atlantic cod macrophages, including miR-214-1-5p, suggesting miR-214 may play an important role in the antiviral immune response (*Eslamloo et al., 2017*).

In short, miR-214 plays an important regulatory role in immune activity and tolerance of DCs, proliferation and differentiation of T cells, and functions of NK cells and macrophages (Table 1). In-depth research on the functions and applications of miR-214 in immune cells may have positive theoretical and practical significances for the prevention and treatment of various diseases, such as organ transplantation, autoimmune diseases, pregnancy injury, immune tolerance and immunosuppression.

## MiR-214 and tumor immunity

MiR-214 plays a key regulatory role in the proliferation, invasion and metastasis of tumor cells (*Yin et al., 2014*). Recently, studies have found that Treg cells play an important role in mediating immune escape of tumor cells and have become important targets for tumor immunotherapy (*Kasinski Andrea & Slack Frank, 2011*; *Ahmetlić et al., 2019*). Interestingly, miR-214 expression is often up-regulated in several tumor cells (such as Gastric cancer *Ji et al., 2019*) and tumor-secreted miR-214 into recipient T cells by microvesicles (*Yin et al., 2014*). When miR-214-3p from microvesicles enters CD4$^+$T cells and down-regulates PTEN, it can inhibit tumor through antagonizing phosphorylase activity, such as tyrosine kinase (*Myers et al., 1998*; *Yang et al., 2008*). In addition, the activities of Treg cells are often

**Table 1 Expressional changes and biological effects of miR-214 in immune and inflammatory responses.**

| miR-214 | Expressional change | Target gene | Biological effect | Reference |
|---|---|---|---|---|
| miR-214-3p | Up | HSP27 | Promote Tregs cell differentiation | Huan et al. (2017) |
| miR-214-3p | Down | β-catenin | Induce DCs tolerance and inhibit ovarian cancer | Gordon et al. (2014), Fändrich (2010) and Gu et al. (2015) |
| miR-214-3p | Up | PTEN | Promote T cell activation and proliferation, inhibit Tregs cell proliferation and tumor growth | Velasco et al. (2006), Jindra et al. (2010), Yin et al. (2014), Yang et al. (2008), Zhang et al. (2010) and Myers et al. (1998) |
| miR-214-3p | Up | CD3 ζ | Inhibit T cell activation | Xiao et al. (2019) |
| miR-214-3p | Down | PDL1 | Regulate T cells and further mediate the immune response of tumor cells | Song, Park & Uhm (2019), Sun, Zhang & Zhang (2019) and Wang et al. (2017) |
| miR-214-3p | Down | B7-H3 | Regulate the polarization pathway of M2 macrophages | Fauci et al. (2012) |
| miR-214-3p | Up | A2AR | Amplify inflammatory effect | Zhao et al. (2015) |
| miR-214-5p | Up | TSG-6 | Promote proinflammatory factor release | Hu et al. (2018) |
| miR-214-3p | Up | TWIST1 | Promote aortic valve stromal cell calcification | Li et al. (2016a) and Li et al. (2016b) |
| miR-214-3p | Up | MyD88 | Increase the secretion of proinflammatory factors and the number of calcified nodules | Zheng et al. (2019) |
| miR-214-3p | Down | Mfn2 | Promote EMT process and bladder wall fibrosis, induce IC development | Lv et al. (2017) |
| miR-214-3p | Up | TLR4 | Inhibit inflammation and cyst growth | Lakhia et al. (2020) |

up-regulated in tumor patients, thereby promoting the development of tumors (*Pandey et al., 2013*; *Alexander & Steffen, 2014*; *Svajger & Rozman, 2014*). If miR-214-3p activity is blocked or down-regulated in microvesicles, then PTEN expression is up-regulated, Treg cells activity is down-regulated, and DCs immune activity is up-regulated, so the tumor growth may be inhibited. For example, *Yin et al. (2014)* used microvesicles to inoculate anti-miR-214 antisense oligonucleotides to tumor mice via tail vein, which significantly inhibited Treg cells proliferation and tumor growth, indicating that the potential application of miR-214 for tumor micro-regulation of Treg cells (*Yin et al., 2014*) (Table 1).

Programmed cell death receptor ligand 1 (PD-L1) interacting with programmed cell death protein 1 (PD-1) can inhibit T cell activation, induce effector T cell apoptosis, finally inhibit tumor immunity , which is important for using immunotherapy to treat tumors (*Dong et al., 1999*; *Ceeraz, Nowak & Noelle, 2013*; *Zhang, Medeiros & Young, 2018*; *Wang et al., 2019a*; *Wang et al., 2019b*). MiR-214-3p can inhibit the expression of PD-L1 by targeting its 3′UTR, but lncRNA urothelial carcinoma associated 1 (UCA1) can up-regulate PD-L1 expression through inhibiting miR-214-3p, which promotes gastric cancer (GC) cell proliferation and migration and inhibits apoptosis, so miR-214-3p can be used as potential new therapeutic target of GC treatment (*Wang et al., 2017*; *Sun, Zhang & Zhang, 2019*; *Song, Park & Uhm, 2019*). In diffuse large b-cell lymphoma (DLBCL), miR-214 also
targets PD-L1 to regulate T cells, and mediates the immune response of tumor cells (*Song, Park & Uhm, 2019*) (Table 1).

Multiple myeloma (MM) is related to macrophage polarization. In MM patients, lncRNA nuclear paraspeckle assembly transcript 1 (NEAT1) and B7-H3 are up-regulated, but miR-214 is significantly down-regulated (*Fauci et al., 2012*; *Wang, Kang & Shan, 2014*; *Mao et al., 2017*). NEAT1 can directly target miR-214, and miR-214 directly binds to B7-H3. If NEAT1 is silenced, miR-214 will inhibit the expression of B7-H3, thus inhibiting the polarization of M2 macrophages by inhibiting JAK2/STAT3 signaling (*Gao et al., 2020*). Therefore, miR-214 plays an important role in the polarization pathway of MM related M2 macrophages (Table 1).

With the deep-going of the research, we believe tumor therapeutic agents including miR-214 will be developed in the future. Therefore, the thorough understanding of the molecular mechanisms of miR-214, the development of new preparations, the molecular mechanism of tumor treatment, and the feasibility of treatment may be issues that need to solve urgently at this stage.

## MiR-214 and inflammation

MiR-214 plays a key regulatory role in promoting inflammation. For example, adenosine A2A receptors (A2ARs) have anti-inflammatory effects, and up-regulation of miR-214 expression in inflammatory cells can inhibit adenosine A2AR expression. Moreover, the decrease of A2AR expression weakens the inhibition of nuclear factor kappa-B (NF-$\kappa$B) through PKA, which promotes the up-regulation of miR-214-3p to amplify the inflammatory response (*Zhao et al., 2015*). In addition, miR-214 also promotes the release of inflammatory factors. For example, Adipose-derived stem cells (ADSCs) inhibit the inflammation of microglia by secreting tumor necrosis factor-inducible gene 6 protein (TSG-6). TSG-6 inhibits the release of pro-inflammatory factors such as IL-1$\beta$, IL-6 and TNF$\alpha$, while *TSG-6* is negatively regulated by miR-214-5p, so miR-214-5p can increase the release of proinflammatory factors by inhibiting TSG-6 (*Hu et al., 2018*). Furthermore, miR-214 plays a key regulatory role in inflammatory response and calcification of human aortic valve interstitial cells (AVICs). M1 macrophages transmit miR-214 to valvular interstitial cells through microvesicles, miR-214-3p directly targets Twist1 and down-regulates it, thus promoting calcification of valve interstitial cells (*Li et al., 2016a*; *Li et al., 2016b*). Further research finds that miR-214 is related to the expression level of MyD88 protein. Up-regulation of miR-214-3p promotes the expression of MyD88 and NF-$\kappa$B, while the up-regulation of MyD88 increases the secretion of pro-inflammatory factors and the number of calcified nodules (*Zheng et al., 2019*). Another, miR-214-3p can be selectively inhibited by 17$\beta$-estradiol (E2) or progesterone (P), while E2 and P can indirectly inhibit apoptosis and inflammation-related gene translation. So it is speculated that E2 and P suppress inflammation by inhibiting miR-214-3p expression (*Herzog et al., 2016*). Besides, miR-214-3p is down- expressed in the post-menopausal women's epithelial-mesenchymal transition (EMT) process and the development of interstitial cystitis (IC), and mitofusin 2 (Mfn2) is the target gene of miR-214-3p, so down-regulation of miR-214-3p promotes the EMT process and bladder wall fibrosis, leading to IC in

postmenopausal women (*Lv et al., 2017*). However, miR-214-3p is up- regulated in the kidney, pancreas and serum of hyperlipidemic pancreatitis (HP) rats. Up- regulation of miR-214-3p inhibits PTEN expression, but increases the level of P-Akt in HP kidneys which may be a possible mechanism for inducing severe symptoms of pancreatitis, and exacerbates HP-induced pathological changes, kidney and pancreas damage, and fibrosis. Therefore, using miR-214-3p as target for the treatment of acute renal injury of HP provides a potential and effective method for the clinic (*Yan et al., 2020*).

Interestingly, miR-214 also plays a regulatory role in inhibiting inflammation. Substantial interstitial inflammation caused by renal cysts is often ignored, and miR-214 is up-regulated in cystic kidney of autosomal dominant polycystic kidney disease (ADPKD) patients (*Lakhia et al., 2020*). The up-regulation mechanism of miR-214 is mainly because of the pro-inflammatory TLR4/IFN-$\gamma$/STAT1 pathways activating the miR-214 host gene (*Watanabe et al., 2008*). In turn, miR-214-3p targets TLR4, and inhibits the inflammatory response. Therefore, the up-regulation of miR-214-3p is a compensatory protective response of the cyst microenvironment, which can inhibit inflammation and cyst growth.

In summary, miR-214 plays different roles in multiple inflammatory response pathways (Table 1). Studying the pro-inflammatory and anti-inflammatory functions of miR-214 can provide positive theoretical basis for understanding the molecular mechanism of inflammatory response and developing new strategies for the diagnosis and treatment of inflammatory diseases.

## MiR-214 and virus

MiRNAs directly target RNA virus genes or affect the replication and pathogenesis of virus through altering the host transcriptome (*Trobaugh & Klimstra, 2017*). It is found that miR-214 is differentially expressed in virus-infected tissues. For example, the expression of miR-214-3p is up-regulated in the plasma and myocardial cells of patients with viral myocarditis (VM) infected with coxsackievirus, but the specific mechanism is still unclear (*Chen et al., 2015*). Coxsackie adenovirus receptor (CAR) protein is an adenovirus receptor, and miR-214-3p can inhibit adenovirus replication by targeting the 3′-untranslated region of early region 1A (E1A) mRNA (*Yanagawa-Matsuda et al., 2012*). Therefore, in-depth study of the mechanism and biological effects of miR-214 in virus-infected tissues may have important theoretical and practical significance for the prevention and treatment of viral diseases including viral myocarditis.

Recent studies have found that miR-214 can inhibit the replication of fish viruses. For instance, miR-214-3p effectively inhibits siniperca chuatsi rhabdovirus (SCRV) replication, which may provide a new approach for the development of effective SCRV infection prevention strategies (*Zhao et al., 2019*). In addition, miR-214-3p also targets the coding regions of viral N and P to inhibit snakehead vesiculovirus (SHVV) replication (*Zhang et al., 2017*). Another study discovers that miR-214-3p can also target glycogen synthase (GS) gene and inhibit SHVV replication, because GS gene is the key gene for SHVV replication. So, miR-214-3p can inhibit SHVV replication from multiple aspects through multiple target genes, which provides several possibilities for the prevention and control of SHVV (*Zhang et al., 2019*).

## MiR-214 and molecular markers

In recent years, the application value of miR-214 as a diagnostic marker has attracted widespread attention.

In inflammatory diseases, miR-214 is one of the three lowest expressed miRNAs in gingival tissue inflammation in Japanese dental patients. It can be determined that abnormal expression of miR-214 is associated with chronic periodontitis, which provides a basis for the diagnosis of periodontal inflammatory diseases (*Ogata et al., 2014*). MiR-214 also is used as a non-invasive biomarker for the diagnosis of ankylosing spondylitis (AS). The expression level of miR-214 in serum of AS patients is significantly lower than that of normal people, and which is significantly correlated with the active C-reactive protein (CRP) of AS disease. Therefore, miR-214 as a diagnostic marker of AS disease provides a powerful help for the treatment and prevention of AS (*Kook et al., 2019*). Up-regulation of STAT6 promotes the secretion of proinflammatory cytokines in intestinal epithelial cells, and then participates in the inflammation response to induce ulcerative colitis (UC) (*Rosen et al., 2011*). *STAT6* is a direct target of miR-214-3p, so targeting *STAT6* pathway by miR-214-3p may become a new therapeutic target for UC (*Li et al., 2017*).

In tumor diseases, the expression of miR-214-3p in the plasma of gastric cancer (GC) patients is significantly higher than that of normal people, and GC patients with high miR-214 expression may have larger tumor lymphatic metastasis and tumor node metastasis (TNM) stage, higher levels of CEA (Carcinoembryonic antigen) and carbohydrate antigen 19-9 (CA19-9), and the survival rate is low. The high sensitivity and specificity of miR-214-3p for GC have high application value in the diagnosis and prognosis of GC (*Zhang et al., 2015*; *Ji et al., 2019*). In addition, miR-214 expression is down-regulated in human cholangiocarcinoma exosomes, sinonasal inverted papilloma (SNIP), difuse large B cell lymphoma (DLBCL) and bladder cancer (BC), which suggests that miR-214 has potential value as a molecular marker and therapeutic target (*Xie et al., 2017*; *Kitdumrongthum et al., 2018*; *Teng et al. 2018*; *Sun, Zhang & Zhang, 2019*). In breast cancer, the proliferation and migration ability of tumor cells with over-expressed miR-214-3p declines, and the cells are induced to apoptosis and interfered with the cell cycle (*Liu et al., 2016*). Similarly, the expression of miR-214-5p also decreases in hepatocellular carcinoma (HCC) tissues and cells. The over-expression of miR-214-5p can decrease cell proliferation, reduce cell migration, and block the cell cycle in G0/G1 phase (*Pang et al., 2018*). Above results indicate that miR-214 plays a key role in inhibiting breast cancer and HCC, and may become a potential biomarker and therapeutic target. MiR-214 is significantly down-regulated in esophageal squamous cell carcinoma (ESCC), and over-expression of miR-214 may impair the invasion and migration ability of Eca109, TE1 and KYSE150 cells. Therefore, miR-214 may have potential application value as a diagnostic marker and therapeutic target of ESCC (*Lu et al., 2016*). In addition, miR-214 expression is down-regulated in colon cancer tissues and cells. MiR-214-3p can inhibit the cell viability and development of colon cancer by inhibiting ADP-ribosylation factor-like protein 2 (ARL2) and mitochondrial transcription factor A (TFAM) (*Long et al., 2015*; *Wu et al., 2018a*; *Wu et al., 2018b*). So, miR-214 may be an important target for the treatment of colon cancer.

In other diseases, miR-214-3p expression is up-regulated in chronic kidney disease. Mitochondrial dysfunction is related to the pathogenesis of chronic kidney disease. MiR-214-3p has a pathogenic role in chronic kidney disease by disrupting mitochondrial oxidative phosphorylation, so miR-214-3p has the potential to become a therapeutic target and diagnostic biomarker for chronic kidney diseases such as nephritis (*Bai et al., 2019*). In addition, 6 miRNAs including miR-214-3p are found to be dysregulated in diabetic kidney disease (DKD), and these miRNAs are involved in the pathogenesis of apoptosis, fibrosis, and accumulation of extracellular matrix related to the pathogenesis of DKD, which indicates that miR-214-3p may have the potential to represent the disease biomarker (*Assmann et al., 2018*). In addition, miR-214-3p is significantly up-regulated in the pathogenesis of myocardial ischemia/reperfusion (I/R) injury, which provides new targets for myocardial I/R damage (*Wang et al., 2016*). MiR-214 also plays an important role in fibrotic diseases. Increasing the expression of miR-214-3p reduces the expression of collagen $\alpha$1 and connective tissue growth factor (CTGF) in endometriosis matrix and endometrial epithelial cells, which provides another treatment for endometrium heterotopic fibrosis (*Wu et al., 2018a*; *Wu et al., 2018b*). Interestingly, miR-214 also plays an important role in musculoskeletal metabolism, bone formation, and other bone diseases. Specifically, miR-214-3p mediates skeletal muscle myogenesis and the proliferation, migration and differentiation of vascular smooth muscle cells. MiR-214-3p also regulates bone formation by targeting specific molecular pathways and expression of various osteoblast-related genes (*Sun et al., 2018*). For example, osteoclast-derived exosome miR-214-3p transferred to osteoblasts can inhibit bone formation (*Li et al., 2016a*; *Li et al., 2016b*). MiR-214's role in primary osteoporosis may be through inhibiting the expression of osterix to inhibit bone formation (*Mohamad et al., 2019*). So miR-214 may be an important potential target for the treatment of bone diseases.

In brief, more and more studies have shown the potential values and application prospects of miR-214 as a diagnostic marker in diseases such as inflammation and tumor (Table 2). It is believed that in the near future, miR-214 will truly appear in clinical practice detection as a diagnostic marker and play its due value for the diagnosis and treatment of clinically relevant diseases.

## CONCLUSIONS

With the deepening of research, the function and mechanism of miR-214 in the fields of immune cell regulation, inflammatory response, tumor immunity and virus replication are gradually revealed. Moreover, the potential clinical application value of miR-214 as a biomarker has attracted increasing attention. According to the research status, miR-214 has promising prospects in the following aspects: Firstly, miR-214 may have in-depth research value in the prevention and treatment of diseases such as organ transplant rejection, autoimmune disease and immune tolerance; Secondly, miR-214 regulates tumor microenvironment to make it have the ability to inhibit the immune escape of tumor cells and its potential application value; Thirdly, abnormal expression of miR-214 can affect the replication of several viruses, which indicates that miR-214 has good development

**Table 2   Biological functions of miR-214 as a molecular marker.**

| miRNA | Related disease | Expressional change | Application prospect | Reference |
|---|---|---|---|---|
| miR-214-3p | Periodontitis | Down | Biomarker for the diagnosis of periodontitis-related diseases | *Ogata et al. (2014)* |
| miR-214 | Ankylosing spondylitis | Down | Non-invasive biomarker for the diagnosis of ankylosing spondylitis | *Kook et al. (2019)* |
| miR-214-3p | Ulcerative colitis | Down | Therapeutic target for ulcerative colitis | *Rosen et al. (2011)* |
| miR-214-3p | Gastric cancer | Up | Biomarker value in the diagnosis and prognosis of gastric cancer | *Zhang et al. (2015)* and *Ji et al. (2019)* |
| miR-214-3p | Cholangiocarcinoma | Down | Biomarker for the diagnosis and treatment of cholangiocarcinoma | *Kitdumrongthum et al. (2018)* |
| miR-214-3p | Bladder cancer | Down | A potential therapeutic target for the treatment of bladder cancer | *Xie et al. (2017)* |
| miR-214-3p | Difuse large B cell lymphoma | Down | Biomarker for good prognosis of difuse large B cell lymphoma | *Sun, Zhang & Zhang (2019)* |
| miR-214-3p | Sinonasal inverted papilloma | Down | Biomarker for the diagnosis and treatment of sinonasal inverted papilloma | *Teng et al. (2018)* |
| miR-214-5p | Hepatocellular carcinoma | Down | Potential biomarker and therapeutic target for HCC | *Pang et al. (2018)* |
| miR-214-3p | Esophageal squamous cell carcinoma | Down | Potential diagnostic marker and therapeutic target of ESCC | *Lu et al. (2016)* |
| miR-214-3p | Colon cancer | Down | Potential target for the treatment of colon cancer | *Long et al. (2015)*, *Wu et al. (2018a)* and *Wu et al. (2018b)* |
| miR-214-3p | Nephritis | Up | Therapeutic target and diagnostic biomarker for chronic kidney disease | *Bai et al. (2019)* |
| miR-214-3p | Diabetic kidney disease | Up | Potential diagnostic marker for diabetic kidney disease | *Assmann et al. (2018)* |
| miR-214-3p | Ischemia/reperfusion | Up | Biomarker for the diagnosis and treatment of myocardial I/R injury prevention | *Wang et al. (2016)* |
| miR-214-3p | Fibrotic diseases | Up | Potential target for treatment of endometrium heterotopic fibrosis | *Wu et al. (2018a)* and *Wu et al. (2018b)* |
| miR-214-3p | bone diseases. | Up | Potential target for the treatment of bone diseases | *Sun et al. (2018)* and *Mohamad et al. (2019)* |

prospect in the diagnosis and treatment of certain viral diseases; Finally, miR-214 has potential application value as a diagnostic marker and therapeutic target in multiple diseases. In short, deep study on the regulatory relationship and molecular regulatory mechanism of miR-214 not only provides important theoretical basis for scientific issues such as immune regulation, tumor treatment, inflammation diagnosis, and antivirus, but also paves the way for actively developing new strategies for the prevention and treatment of these diseases. It is believed that miR-214 will have great research value and bright application prospects whether it is used as a drug target for disease treatment or as a molecular marker for disease diagnosis and prognosis.

### Funding

This work was supported by the Science Foundation of Heilongjiang Province [grant number LH2019C073], and the Postgraduate Innovation Project of Harbin Normal University [grant number HSDSSCX2020-07]. The funders had no role in study design, data collection and analysis, decision to publish, or preparation of the manuscript.

### Grant Disclosures

The following grant information was disclosed by the authors:
Science Foundation of Heilongjiang Province: LH2019C073.
Postgraduate Innovation Project of Harbin Normal University: HSDSSCX2020-07.

### Competing Interests

The authors declare there are no competing interests.

### Author Contributions

- Qiuyuan Wang conceived and designed the experiments, performed the experiments, analyzed the data, prepared figures and/or tables, and approved the final draft.
- Yang Liu performed the experiments, prepared figures and/or tables, and approved the final draft.
- Yiru Wu performed the experiments, prepared figures and/or tables, and approved the final draft.
- Jie Wen performed the experiments, prepared figures and/or tables, and approved the final draft.
- Chaolai Man performed the experiments, authored or reviewed drafts of the paper, and approved the final draft.

### Data Availability

This is a literature review; there is no data.

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
