# Peer review of "Immune function of miR-214 and its application prospects as molecular marker"

_PeerJ, doi:10.7717/peerj.10924_

## Round 0.1 · original submission · Major Revisions

Please address the concerns point by point in your resubmission as the reviewer's comments are meant to improve your manuscript.

Reviewer 1 ·

Basic reporting

I think the review is not broad since it is only about one microRNA.

I think the review is not cross-disciplinary since it is mainly about immunology.

However, the review is long enough as is, and it might not improve by increasing its scope.

To my knowledge this topic has not been reviewed recently.

Experimental design

no comment

Validity of the findings

Overall, this review provides a catalog of published findings, but the authors do not synthesize a refreshing point of view.

Additional comments

My brief comments and suggestions:

1) The text could benefit from professional editing by a native English speaker. I’ve highlighted a few examples (not exhaustively) in attached, annotated PDF where the English is awkward and could be improved for clarity.

2) Authors should provide more details to their literature Survey Methodology. For example, what exclusion criteria were used? Their PubMed search must have yielded many articles. How did authors choose which ones to cite in their References list? In particular, PeerJ is concerned about selection bias.

3) Since, “miR-214 is located in the dynamin-3 gene…,” does its expression mimic that of its host gene dynamin-3 (perhaps because its transcription is regulated by a shared enhancer)?
Since miR-199a is located right next to miR-214, is it known whether they functionally cooperate (eg. in analogy to miR-17 cluster).

Based on miRbase (http://www.mirbase.org/cgi-bin/mirna_entry.pl?acc=MI0000290 ), in humans, miR-214-5p is hardly expressed based on 136 published RNA-seq experiments. Perhaps this is an important detail to share with readers. Also, this calls into question the utility of two findings listed in Table 2:

Kook HY et al. (2019) miR-214-5p is down in Ankylosing Spondylitis
Pang J et al. (2018) miR-214-5p is down in Hepatocellular Carcinoma
Could the authors please check the primary data regarding expression of miR-214-5p in those two papers? If the expression of miR-214-5p is low to begin with, it might not be so trivial to assay further down-regulation in a reliable fashion. Is expression of miR-214-3p also down-regulated? If not, why would that be?

4) The central image in Fig 1 looks like it’s copied from miRbase (http://www.mirbase.org/cgi-bin/mirna_entry.pl?acc=MI0000290 ). If so, miRbase should be cited appropriately.

5) In the Conclusion section, PeerJ requests authors to identify unresolved questions and gaps in knowledge. For example, is there a need to systematically identify direct targets of miR-214?

Annotated reviews are not available for download in order to protect the identity of reviewers who chose to remain anonymous.

Reviewer 2 ·

Basic reporting

This is a comprehensive review of the literature pertaining to miR-214. It seems well organized and informative.

Experimental design

no comment

Validity of the findings

no comment

Reviewer 3 ·

Basic reporting

In the review article “Immune function of miR-214 and its application prospect as molecular marker” the authors compiled the available data related to miR-214 and its role in immune function.
This review is not at all well written. The literature review is written in pieces and is not well connected with each other. It looks more like notes rather than a review article. There is an absence of critical analysis of the available data from the literature. The review lacks flow and jumps from one information to the other throughout the article which makes it difficult for the reader to focus and understand. The language needs a lot of improvement as sentences are too long and confusing making it difficult for the reader to understand the gist of the review. There are many redundant sentences. The authors have used a lot of incorrect scientific terminology. In some instances, authors have extrapolated interpretation without any scientific evidence.

Experimental design

The review is not well organized. Looks like notes of the available literature.

Validity of the findings

NA

Additional comments

For review articles, authors should summarize and analyze the scientific findings and critically discuss the outcome from their perspective which is lacking in this review. Language is an issue that makes it difficult to read and understand the review and many incorrect scientific terms are used.
Few examples are
Confusing sentences: 49-54, 114-115
Wrong use of scientific terms:
Line 129: damage anti-tumor immunity
Line 107: what is microbubble structure?
Line 113: what are microcapsules?
Overinterpretation of results
Line 147: how miR-124 plays a role in the polarization pathway of M2 macrophages?
Role of miR-124 in organ transplantation
Line 31: Consistency in the use of the miR-214 or MIR-214
Line 217: Glycogen is repeated twice.

---

## Round 0.2 · Minor Revisions

Your changes are acceptable but there are still some problems with English editing (e.g. researches is not a word). Please have some type of editing service go through the manuscript and make the needed grammatical changes. Once this is done, I will not resend the manuscript to reviewers as it will be accepted.

---

## Round 0.3 · accepted · Accept

Your changes have been accepted and the manuscript is now accepted for publication.